# FedNAMs: Performing Interpretability Analysis in Federated Learning Context

## Abstract

Federated learning continues to evolve but faces challenges in interpretability and explainability. To address these challenges, we introduce a creative approach employing Neural Additive Models (NAMs) within a federated learning framework. These new Federated Neural Additive Models (FedNAMs) approach merges the advantages of NAMs, where individual networks concentrate on specific input features, with the decentralized approach of federated learning, ultimately producing interpretable analysis results. This integration enhances privacy by training on local data across multiple devices, thereby minimizing the risks of data centralization and enhancing model robustness and generalizability. FedNAMs maintain detailed feature-specific learning, making them especially valuable in sectors like finance and healthcare. They facilitate training client-specific models to integrate local updates, preserve privacy, and reduce centralization concerns. Our studies on various text and image classification tasks, using datasets such as OpenFetch ML Wine, UCI Heart Disease, and Iris, show that FedNAMs deliver strong interpretability with minimal accuracy loss compared to traditional Federated Deep Neural Networks (DNNs). The research involves notable findings, including the identification of critical predictive features at the client level as well as at the global level. Volatile acidity, sulfates, and chlorides for wine quality. Chest pain type, maximum heart rate, and number of vessels for heart disease. Petal length and width for iris classification. This approach strengthens privacy and model efficiency and improves interpretability and robustness across diverse datasets. Finally, FedNAMs generate insights on causes of highly and low interpretable features.

## 1 Introduction

Deep neural networks (DNN) have delivered remarkable results in areas like computer vision (Himeur et al., 2023) and language modeling (Che et al., 2023). While understanding the mechanisms behind their predictions remains challenging, leading them to be often regarded as black-box models. This lack of interpretability limits their use in critical fields such as finance, criminal justice, and healthcare. Various efforts have been made to clarify the predictions made by deep neural networks (DNNs) in Federated learning environments. For instance, a class of methods, exemplified by LIME (Ribeiro et al., 2016), seeks to explain individual predictions by locally approximating the neural network with interpretable models, such as linear models and shallow decision trees for each client in the federated learning environment. However, these methods frequently fall short in robustness and comprehensive understanding of the model, and their explanations may not accurately reflect the computations of the original model or lack the detail necessary to grasp the model's behavior (Zhang & Li, 2023) fully. In this research, we propose FedNAMs, an interpretable federated learning framework based on Neural Additive Models (NAMs) (Agarwal et al., 2021). We compare the performance and interpretability of this framework to traditional federated learning models. Furthermore, the study explores the trade-offs between interpretability and predictive accuracy in a federated environment.

Interpretable Federated Learning (IFL) has emerged as a promising technology to enhance system safety robustness and build trust among FL stakeholders, drawing considerable research interest from academia and industry in recent years (Li et al., 2023). In contrast to existing interpretable AI methods developed for centralized machine learning, IFL presents more significant challenges due to enterprises' limited access to local data and the constraints imposed by local computational and communication resources. IFL is inherently interdisciplinary, requiring expertise in machine learning,

optimization, cryptography, and human factors to devise effective solutions. This complexity makes it challenging for new researchers to stay abreast of the latest developments. A comprehensive survey paper on this critical and rapidly evolving field has yet to exist. Federated Learning (FL) is a groundbreaking approach to machine learning that enables models to be trained on decentralized data sources while safeguarding data privacy. This method is especially advantageous in healthcare, finance, and mobile applications, where sensitive data is distributed across multiple locations (Aouedi et al., 2022). Traditional centralized learning approaches present significant privacy risks and are often impractical due to data transfer limitations and regulatory constraints. Despite the benefits of FL, a key challenge persists in the interpretability of the models it produces (Zhang et al., 2024b). Most FL models, particularly those based on deep learning, operate as black boxes, offering minimal insight into their decision-making processes. This lack of transparency impedes their adoption in critical fields where understanding the reasoning behind model predictions is crucial. The current federated learning landscape is dominated by complex, opaque models that, although highly accurate, provide little transparency. There is an increasing demand for interpretable machine learning models to elucidate their inner workings and decision-making processes. Neural Additive Models (NAMs) (Agarwal et al., 2021), which combine the robustness of neural networks with the interpretability of additive models, represent a promising solution. However, integrating NAMs into the federated learning framework presents significant challenges, including maintaining interpretability across distributed nodes and ensuring overall model performance.

In this research paper, we propose a federated learning framework while imposing specific constraints on the architecture of neural networks using interpretable models known as Neural Additive Models (NAMs). While implementing tabular data, these glass-box models maintain a high level of interpretability with minimal loss in prediction accuracy. NAMs are part of the Generalized Additive Models (GAMs) family (Hastie, 2017), which takes the form:

$$g(E[y]) = \beta + f_1(x_1) + f_2(x_2) + \cdots + f_K(x_K) \tag{1}$$

where $x = (x_1, x_2, \ldots, x_K)$ represents the input with $K$ features, $y$ is the target variable, $g(\cdot)$ is the link function, and each $f_i$ is a univariate shape function with $E[f_i] = 0$.

In traditional GAMs, the model fitting uses the analytical method of iterative back fitting with smooth low-order splines that effectively reduce overfitting. While more recent GAMs (Hastie, 2017) use boosted decision trees to enhance accuracy and allow the learning of abrupt changes in the feature-shaping functions. Hence, it captures better patterns in actual data that smooth splines struggle to model. This paper explores the use of deep neural networks (DNNs) to fit generalized additive models (NAMs) in a federated learning setup. NAMs provide interpretable insights on DNNs, which is essential for federated learning as models will be more understandable across multiple decentralized nodes. Unlike tree-based GAMs, NAMs can adapt to multiclass, multitask, or multi-label learning. In a federated learning scenario, models are trained efficiently across distributed nodes using shared resources. Therefore, FedNAMs will be more scalable than the traditional GAMs.

## 2 BACKGROUND AND EXISTING WORKS

Federated Learning (FL) (McMahan et al., 2017), (Liu et al., 2024), (Balija et al., 2024), (Hard et al., 2018) is a machine learning paradigm designed to train models across multiple decentralized devices or servers while preserving data privacy. Unlike traditional centralized learning approaches, where data is aggregated and processed in a central location, FL allows data to remain localized while only sharing model updates. This approach is particularly beneficial in domains where data privacy and security are paramount, such as healthcare, finance, and mobile applications. Neural Additive Models (NAMs) (Agarwal et al., 2021) are machine learning models that combine the flexibility and power of neural networks with the interpretability of additive models. NAMs decompose the prediction task into individual functions, each contributing to the final prediction transparently. This decomposition facilitates a clearer understanding of how different features influence model'sel's predictions, addressing the interpretability challenge inherent in traditional neural networks.

Federated learning has garnered significant attention in recent years, leading to the development of various frameworks and methodologies to enhance its effectiveness and efficiency. McMahan et al. (McMahan et al., 2017) introduced the concept of Federated Averaging (FedAvg), a fundamental

algorithm in FL that aggregates model updates from multiple clients to create a global model. Subsequent research has focused on improving the robustness and scalability of FL systems. Bonawitz et al. (Bonawitz, 2019) explored secure aggregation techniques to ensure privacy-preserving model updates, while Kairouz et al. (Kairouz et al., 2021) provided a comprehensive survey of FL advancements, highlighting the challenges and opportunities in the field. Interpretability has become a critical aspect of machine learning, especially in applications requiring transparency and trust. Ribeiro et al. (Ribeiro et al., 2016) introduced LIME (Local Interpretable Model-agnostic Explanations), a method to interpret predictions of any classifier by approximating it with an interpretable model locally. Shapley values, derived from cooperative game theory, have also been employed to attribute contributions of individual features to model predictions, as seen in the work by Lundberg and Lee (Lundberg & Lee, 2017) on SHAP (Shapley Additive explanations). NAMs proposed by (Agarwal et al., 2021) is a novel approach for achieving high predictive accuracy and interpretability. By leveraging the structure of Generalized Additive Models (GAMs) and the learning capabilities of neural networks, NAMs enable transparent and robust predictive models. The individual contributions of features are modeled using neural networks, allowing non-linear relationships while maintaining additive interpretability. Integrating interpretability into federated learning is an emerging research area. Studies have begun exploring combining interpretable models with FL to ensure privacy and transparency. For instance, (Zhang et al., 2024a) proposed FedGNN, a federated learning framework using Graph Neural Networks emphasizing interpretability. Another approach by Gu et al. (Gu et al., 2021) introduced interpretable FL by incorporating inherently interpretable decision trees into the FL framework.

## 3 NEURAL ADDITIVE MODELS

Neural Additive Models (NAMs) are a class of machine learning models that combine the flexibility of neural networks with the interpretability of additive models. NAMs have gained attention for their ability to provide accurate predictions while enabling human-understandable insights into how the model makes its predictions. NAMs incorporate a series of neural network layers to a Generalized Additive Model (GAM) (Hastie, 2017). The neural network layers allow the model to capture complex interactions between variables, while the GAM component provides an interpretable baseline model. NAMs can be used for classification and regression tasks and trained using standard optimization techniques. Compared with other methods for interpreting black-box models, NAMs provide more detailed and faithful explanations of the model's behavior. Therefore, they are beneficial in high-stakes domains such as healthcare, finance, and criminal justice. It is essential to understand how a model makes its predictions. NAMs leverage innovative ExU hidden units, enabling sub-networks to learn the more linear functions crucial for accurate additive models. By forming an ensemble of these networks, NAMs can provide uncertainty estimates, enhance accuracy, and mitigate the high variance that may arise from enforcing a highly linear learning process. We employed an NAM architecture consisting of three hidden layers containing 20 neurons. During training, the model learns the weights between the input features and the neurons in each layer, optimizing network'srk's ability to capture linear and non-linear relationships in the data.

## 4 PROBLEM FORMULATION

Our proposed architecture, which adapts Neural Additive Models (NAMs) for a federated learning context, is designed to balance interpretability and accuracy. It addresses a network optimization problem focused on uncovering the relationships between input features and the output. In this architecture, each input feature is processed by an individual neural network, resulting in a model that maintains this delicate balance. By maintaining separate neural networks for each feature, this approach preserves the interpretability inherent in additive models while harnessing the representational strength of neural networks to achieve higher predictive performance.

$$w^{t+1} \leftarrow \sum_{client_i=1}^{K} \frac{n_i}{n} w_{client_i}^{t+1} \qquad (2)$$

where $w^{t+1}$ is the global model at iteration $t+1$ and shows the update rule where $w^{t+1}_{client_i}$ is the weighted sum of the clients model.

$$f_1(x_{1_{\text{final}}}) = \frac{f_{11}(x_1) + f_{21}(x_1) + f_{31}(x_1) + \cdots + f_{n1}(x_1)}{n} \tag{3}$$

$$f_1(x_{2_{\text{final}}}) = \frac{f_{12}(x_2) + f_{22}(x_2) + f_{32}(x_2) + \cdots + f_{n2}(x_2)}{n} \tag{4}$$

$$f_1(x_{3_{\text{final}}}) = \frac{f_{13}(x_3) + f_{23}(x_3) + f_{33}(x_3) + \cdots + f_{n3}(x_3)}{n} \tag{5}$$

$$\vdots \tag{6}$$

$$f_1(x_{k_{\text{final}}}) = \frac{\sum_{i=1}^{n} f_{i1}(x_k)}{n} \tag{7}$$

where $f_1(x_{1_{\text{final}}})$ is the final aggregated function for input features $x_1$, which is the sum of the subfunctions $f_{i1}(x_1)$ from each client (for $i = 1, 2, \cdots, n$), divided by the total number of clients $n$. This indicates that each feature function is learned separately across different clients, and their contributions are averaged to produce the final function of that feature. The $g(E[y_{\text{client1}}])$ represents the expected prediction for client $i$. Figure 1 shows the neural additive model architecture and two different neural networks considered for text and image datasets in Figure 2.

$$g(E[y_{\text{client1}}]) = \beta + f_{11}(x_1) + f_{12}(x_2) + \cdots + f_{1K}(x_K) \tag{8}$$
$$g(E[y_{\text{client2}}]) = \beta + f_{21}(x_1) + f_{22}(x_2) + \cdots + f_{2K}(x_K) \tag{9}$$
$$g(E[y_{\text{client3}}]) = \beta + f_{31}(x_1) + f_{32}(x_2) + \cdots + f_{3K}(x_K) \tag{10}$$
$$g(E[y_{\text{client4}}]) = \beta + f_{41}(x_1) + f_{42}(x_2) + \cdots + f_{4K}(x_K) \tag{11}$$

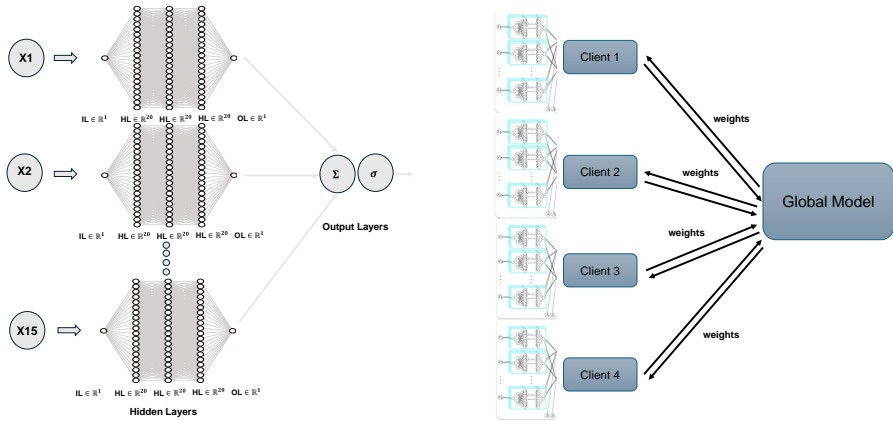

Figure 1: Neural additive model architecture.

## 5 DATASETS

The UCI Heart Disease, OpenML Wine, and Iris datasets are widely recognized benchmarks in machine learning, frequently used for classification tasks across various domains. The UCI Heart Disease dataset contains 1025 instances and 14 patient medical profile attributes. The attributes include demographic and clinical factors such as age, chest pain type, resting blood pressure, serum cholesterol in mg/dl, fasting blood sugar, resting electrocardiographic results (values 0,1,2), maximum heart rate achieved, exercise-induced angina, ST depression induced by exercise relative to rest, the slope of the peak exercise ST segment, number of major vessels (0-3) colored by fluoroscopy, "thal": 0 = normal; 1 = fixed defect; 2 = reversible defect. The primary goal is to predict the presence (1)

Figure 2: Two different neural networks considered for text and image datasets.

or absence (0) of heart disease in patients, making it a valuable resource for research in medical diagnostics. The Wine dataset consists of red variants of the Portuguese wine. The dataset has 1599 instances and 11 attributes such as fixed acidity, volatile acidity, citric acid, residual sugar, chlorides, free sulfur dioxide, total sulfur dioxide, density, pH, sulfates, and alcohol. The Iris dataset is one of the most well-known datasets in machine learning, consisting of 150 instances of iris flowers. Each instance is described by four attributes: sepal length, sepal width, petal length, and petal width. The Iris dataset target variable has three classes corresponding to the three species of iris flowers: Iris-setosa and Iris-versicolor. This dataset is ideal for testing algorithms and visualization techniques due to its simplicity and effectiveness in demonstrating basic classification concepts.

## 6 EXPERIMENTATION AND RESULTS

In this research, we developed a federated learning framework that leverages a standard neural network model and Neural Additive Models (NAMS) to identify both high and low contributing features for each client. For experimentation, we considered three clients in a federated setup. Three datasets used in this setup first go through the preprocessing by scaling features and converting the target to a binary classification model for the UCI Heart Disease and Wine dataset while multi-label classification for the Iris dataset. The dataset is split into training and testing sets and divided into three distinct clients, each receiving a portion of the training data. Each client is trained using the NAM model, which consists of several FeatureNN modules, one for each feature, allowing individual feature contributions to be learned interpretably. The NAM model concatenates outputs from the feature-specific neural networks and passes them through a final output layer for classification. The framework utilizes a robust mechanism of performing hyperparameter tuning for dropouts, learning rate, number of hidden layers in the network, and batch size using grid search across the three clients. Training incorporates early stopping and learning rate scheduling to prevent overfitting and adapt learning rates throughout training. Custom weight initialization using Xavier uniform distribution is applied during training to improve convergence. Furthermore, early stopping is implemented to halt training. Model equations representing each client's specific feature contributions are derived, providing interpretability by highlighting the most and least significant features. Finally, model performance is evaluated based on classification accuracy and metrics such as the ROC-AUC score, with the best hyperparameters being selected based on validation accuracy across all clients.

### 6.1 INTERPRETATION OF FEATURE RELATIONSHIPS

Figure 3 shows images depicting the output variation to different features for the heart dataset. Table 1 and Table 2 represent client-wise feature contributions for UCI Heart disease data and feature attribution values of Captum for UCI Heart disease data, respectively. We benchmarked our framework performance with PyTorch Captum. Our framework offers more detailed and feature-specific interpretability than Captum, which typically provides aggregate feature importance values. Captum generates average attributions for each feature across the entire model, which can obscure individual features' contributions at different learning stages. In contrast, our approach extracts interpretability at multiple stages of the model by independently evaluating the contribution of each feature through specialized sub-networks of NAMs. Figure 4 shows the high and low interpretable features and their causes shown for the heart disease dataset. The plots generated for the Heart

Disease dataset visually represent the relationship between various features and the predicted output for different clients. For instance, the feature x_age demonstrates varying trends across clients, with some models showing a positive correlation between age and the predicted outcome. In contrast, others display an adverse or fluctuating relationship. This suggests that age may have a different impact on the heart disease prediction model for various clients, possibly due to variations in the data distribution or the model's sensitivity to age-related factors. Similarly, the x_cp (chest pain type) feature shows a distinct pattern across clients, where the impact on the model's prediction varies. In some cases, higher values of x_cp increase the predicted output, indicating a higher likelihood of heart disease, while in others, the effect is less pronounced or even reversed. These differences highlight the importance of personalizing models based on specific client data, as the same feature may have differing implications depending on an individual's overall health profile and other contributing factors. Detailed result is shown in Appendix A.

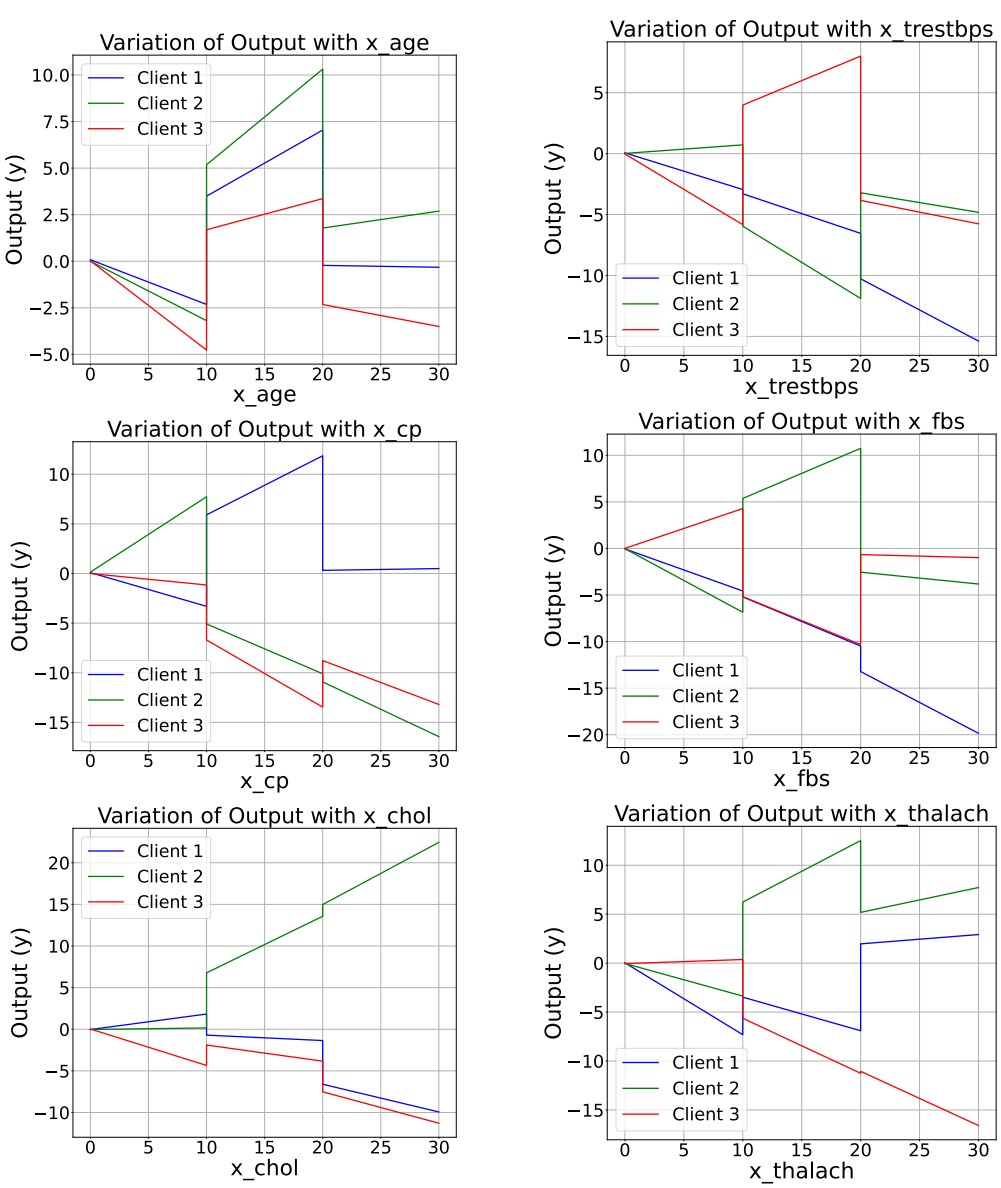

Figure 3: Image depicting the output variation to different features for the heart disease dataset.

| Feature | Client 1 | Client 2 | Client 3 |
|---------|----------|----------|----------|
| thalach | 4.489 | 5.226 | 3.375 |
| thal | 4.360 | 3.416 | 4.298 |
| age | 4.096 | 3.649 | 3.364 |
| ca | 3.838 | 4.041 | 4.246 |
| cp | 3.679 | 4.684 | 3.260 |
| sex | 3.583 | 3.649 | 4.629 |
| trestbps | 3.557 | 3.832 | 3.797 |
| oldpeak | 3.385 | 4.423 | 4.195 |
| fbs | 3.373 | 2.613 | 2.951 |
| restecg | 3.253 | 3.704 | 3.281 |
| exang | 2.926 | 3.928 | 3.626 |
| slope | 2.778 | 2.704 | 3.264 |
| chol | 2.181 | 3.735 | 3.564 |

Table 1: Client-wise feature contributions for UCI Heart disease data.

| Feature | Average Attribution |
|---------|---------------------|
| age | -0.003673 |
| sex | -0.000434 |
| cp | -0.004202 |
| trestbps | -0.002589 |
| chol | -0.000223 |
| fbs | -0.001079 |
| restecg | -0.001987 |
| thalach | -0.004438 |
| exang | 0.003228 |
| oldpeak | -0.010129 |
| slope | -0.004840 |
| ca | 0.001944 |
| thal | -0.008827 |

Table 2: Feature attribution values of Captum (Benchmark) for UCI Heart Disease data.

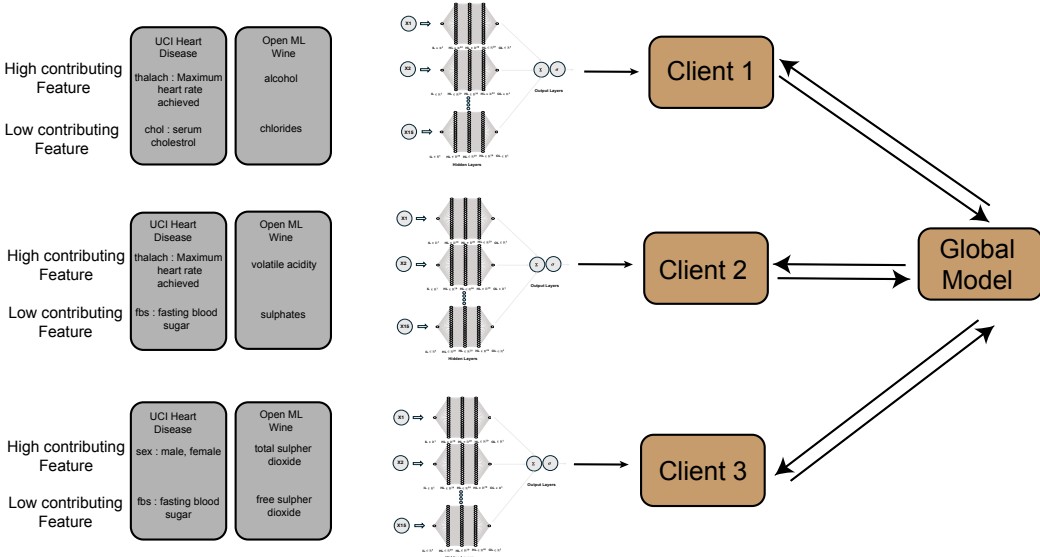

Figure 4: High and low interpretable features and their causes are shown for the heart disease dataset.

## 6.2 INSIGHTS ON FEATURE IMPACTS

Table 3 shows the client-wise feature contributions for the UCI-wine dataset. Figure 5 shows the image depicting the output variation concerning different features of the Iris dataset. Figure 6 shows the benchmark comparison with Meta's Captum (right) for highly contributing pixels (masked) on MNIST data test image. The vertical plots for selected features in the Heart Disease dataset reveal how specific attributes influence model predictions across different clients. For example, the x_trestbps (resting blood pressure) feature shows varying effects: one client's model indicates a sharp increase in predicted risk with higher blood pressure, while another shows a minimal impact. This suggests that resting blood pressure is a significant predictor for some clients but not others. Similarly, x_thalach (maximum heart rate achieved) exhibits diverse influences, with higher heart rates strongly associated with increased heart disease risk in some clients but not others. These variations highlight the importance of assessing feature impact within the context of client-specific data. The analysis of features like x_fixed_acidity and x_volatile_acidity across different clients

shows consistent influence, though the magnitude and direction may vary slightly, suggesting a need for tailored model adjustments. Detailed result is shown in Appendix A.

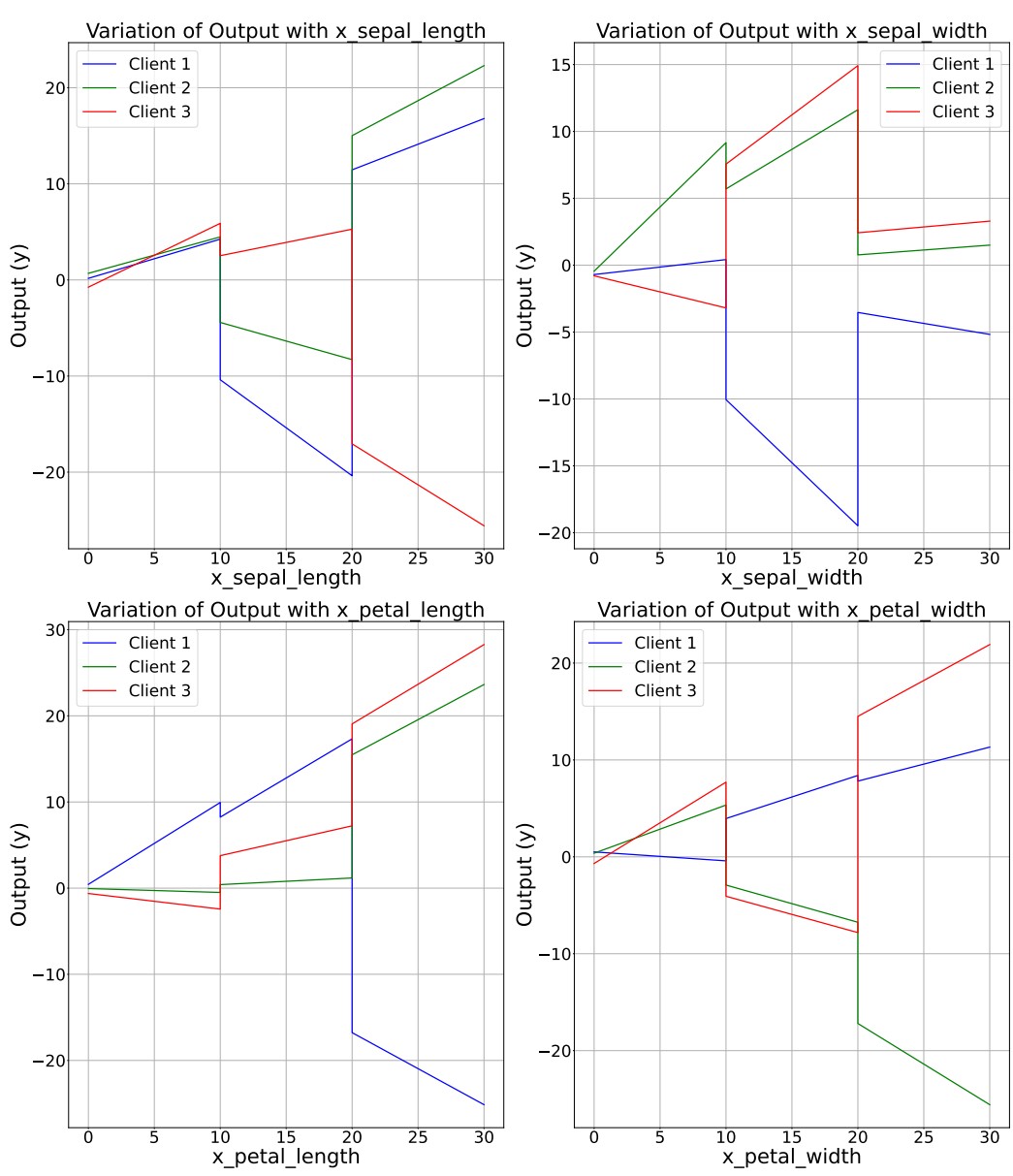

Figure 5: Image depicting variation of output with respect to different features for Iris dataset

The analysis highlights that while each feature consistently impacts model output across different clients, the magnitude and direction of this influence can vary, suggesting the need for client-specific adjustments. For example, x_fixed_acidity shows both positive and negative effects for Client 1, while Client 2 experiences consistent impacts. Features like x_volatile_acidity and x_sulphates significantly affect outcomes with varied client slopes, though the overall patterns are similar. Other features such as x_citric_acid, x_residual_sugar, and x_chlorides display consistent trends with minor variations. Additionally, the comparison of digit '9' images shows how the model emphasizes specific pixel regions (highlighted in black) crucial for accurate predictions, contrasting with the less significant areas in gray. This visualization offers insight into the neural network's interpretability and decision-making process.

| Feature | Client 1 | Client 2 | Client 3 | Meta Captum Average Attribution |
|---|---|---|---|---|
| thalach | 4.489 | 5.226 | 3.375 | -0.004438 |
| thal | 4.360 | 3.416 | 4.298 | -0.008827 |
| age | 4.096 | 3.649 | 3.364 | -0.003673 |
| ca | 3.838 | 4.041 | 4.246 | 0.001944 |
| cp | 3.679 | 4.684 | 3.260 | -0.004202 |
| sex | 3.583 | 3.649 | 4.629 | -0.000434 |
| trestbps | 3.557 | 3.832 | 3.797 | -0.002589 |
| oldpeak | 3.385 | 4.423 | 4.195 | -0.010129 |
| fbs | 3.373 | 2.613 | 2.951 | -0.001079 |
| restecg | 3.253 | 3.704 | 3.281 | -0.001987 |
| exang | 2.926 | 3.928 | 3.626 | 0.003228 |
| slope | 2.778 | 2.704 | 3.264 | -0.004840 |
| chol | 2.181 | 3.735 | 3.564 | -0.000223 |

Table 3: Client-wise feature contributions and Feature attribution values of Captum for UCI Wine dataset with reduced precision.

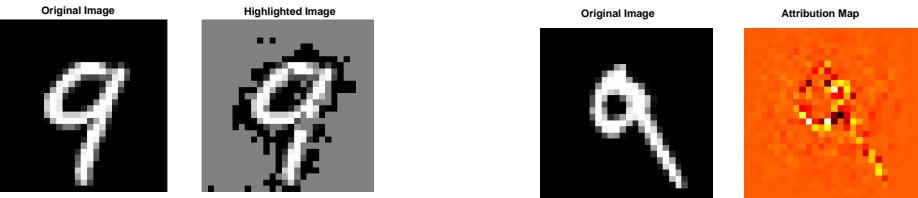

Figure 6: Benchmark comparison with Meta's captum (right) for highly contributing pixels (masked) on MNIST data test image

## 7    CONCLUSION AND FUTURE WORK

This work presents a novel framework for Federated Neural Additive Models (FedNAMs) using Neural Additive Models, an innovative subfamily of Generalized Additive Models (GAMs) designed to leverage deep learning techniques for scalability across large datasets and high-dimensional features. Our approach addresses critical challenges associated with scalability and performance in federated learning, all while maintaining the interpretability that GAMs are known for, distinguishing it from traditional black-box deep neural networks (DNNs). Experiments on various datasets, including the UCI Heart Disease, OpenML Wine, and Iris datasets, demonstrated that FedNAMs achieve state-of-the-art performance across diverse tasks. Despite their smaller and faster architecture than other neural-based GAMs, FedNAMs effectively capture the nuances of federated learning environments, where data is distributed across multiple clients. The observed plot confirms that the heart disease rate increases with age, aligning with real-life data and trends. This validates the correlation between age and heart disease in practical scenarios. Our results reveal that while the models trained on different clients, such as those using the UCI Heart Disease, OpenML Wine, and Iris datasets, exhibit consistent feature contributions, the local data characteristics still influence specific parameter values. This finding is crucial, as it suggests that FedNAMs can maintain personalization at the client level while ensuring generalizability across the entire federated learning system. Future research will focus on further enhancing the scalability and efficiency of federated NAMs, especially in scenarios with a larger number of clients and more complex data distributions. Additionally, efforts will be directed toward performing interpretability analysis in large language models (LLMs) to better understand the decision-making processes of these models in federated environments.

ACKNOWLEDGMENTS

We thank all the reviewers and mentors who provided valuable insights into our work.

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

# APPENDIX

# A    RESULTS

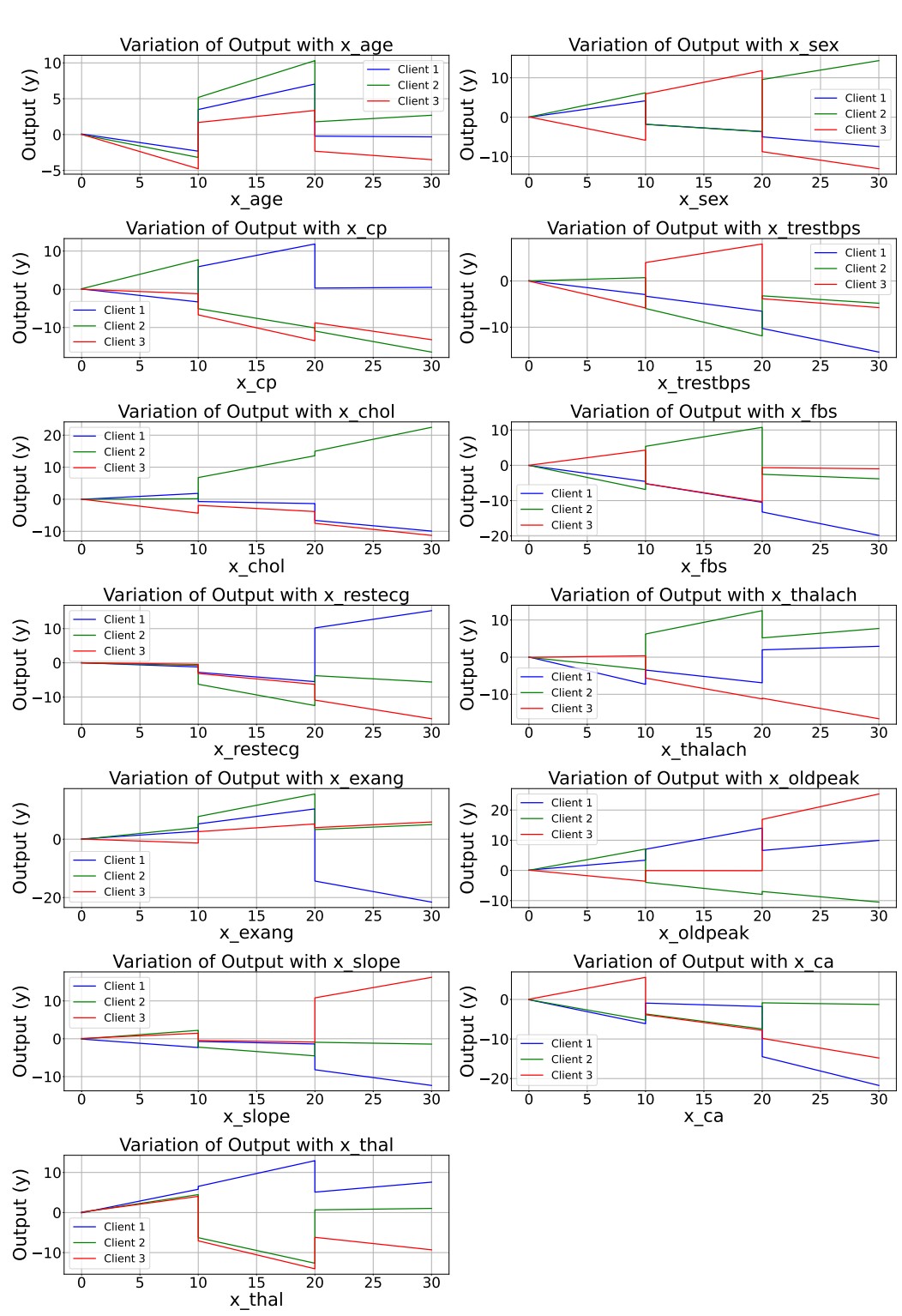

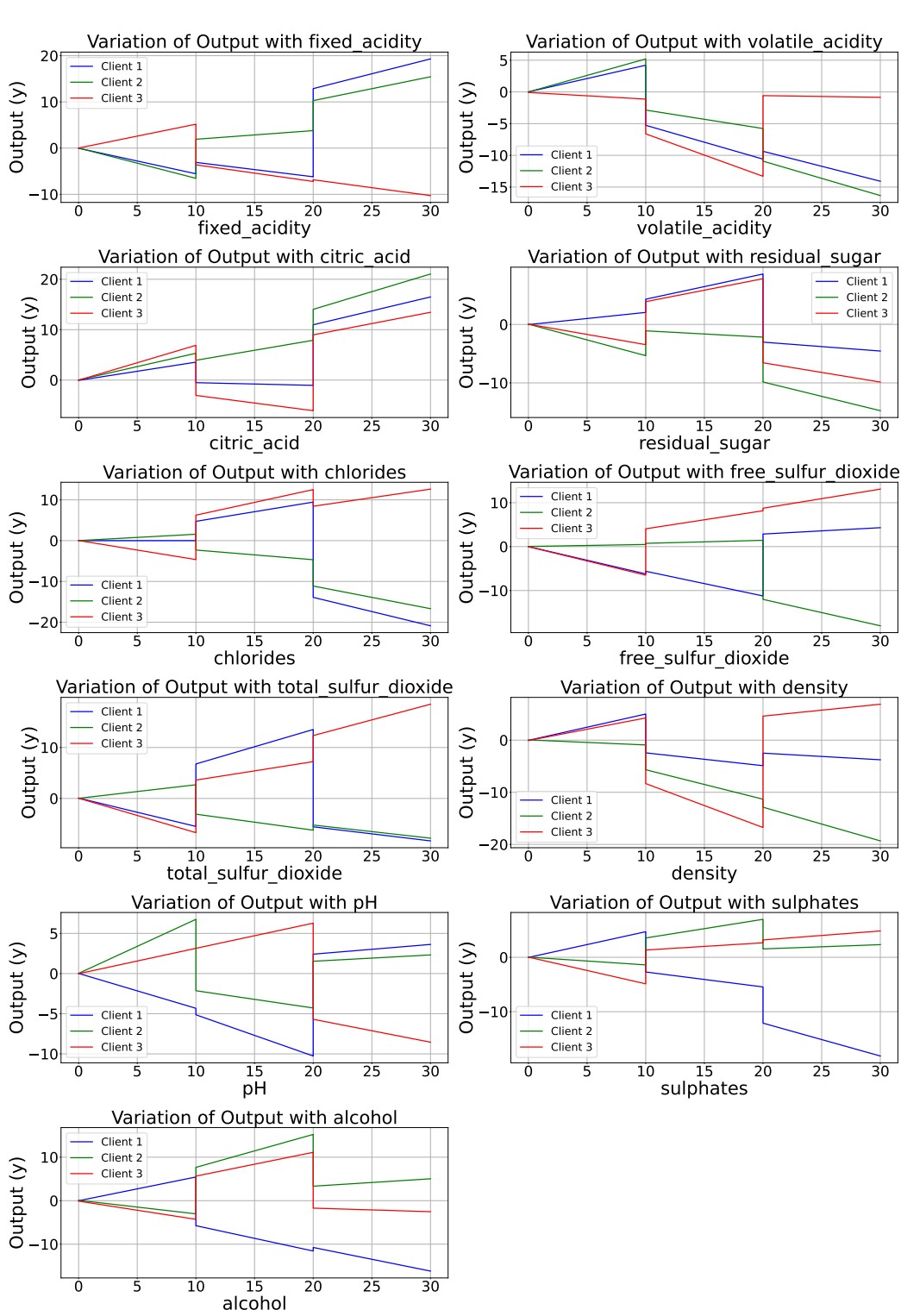

