# OpenReview forum: "Performing Interpretability Analysis in Federated Learning Context"
_ICLR.cc/2025/Conference — ICLR 2025 Conference Withdrawn Submission_

### Official Review · Reviewer_Rwzc · 2024-10-15

**Soundness:** 1
**Presentation:** 1
**Contribution:** 1
**Rating:** 1
**Confidence:** 4

**Summary:**

The authors apply Neural Additive Models (NAMs) in a Federated Learning context. By construction, NAMs are i) more interpretable than Deep Neural Networks (DNNs), as their decision-making process is more transparent ii) more flexible than other Generalized Additive Models (GAMs), due to the ability to train using back-propagation.

The model is trained using the FedAvg paradigm, but instead of one model, one model per feature is trained locally and then averaged.

The experiments consist of studying the feature contributions after training on binarized versions of UCI Heart Disease and Kaggle Wine Reviews, and the Iris dataset.

**Strengths:**

The main strength of the paper is the fact the authors have identified a class of models (NAMs) that (as far as I can tell) has not been extended to the Federated Learning setting yet. Providing interpretable models is very desirable in every area of Machine Learning and Federated Learning is no exception, so NAMs are potentially an interesting approach in that regard.

**Weaknesses:**

* Poor presentation: There are numerous typos and grammatical errors. Some examples are: line 014 (typo), lines 036-037 (grammar error), lines 104,146 (glaring typos), lines 111,114,117 (incorrect citation styles). I advise the authors perform a thorough re-write of the paper, apart from the many errors there are multiple redundant sentences that do not advance the main points of the study. Some statements in the conclusion are incongruent with the paper: "Experiments on various datasets, including the
UCI Heart Disease, OpenML Wine, and Iris datasets, demonstrated that FedNAMs achieve state- of-the-art performance across diverse tasks" when no accuracy numbers have been presented, and there are no baselines that have been compared with.

* Insufficient novelty: The authors have applied the paradigm of FedAvg to NAMs. This is very straight-forward to do, and required no sophistication; instead of averaging one model, K models (where K is the number of features) have been averaged. I do not see why this necessitates a paper. The authors should re-evaluate the motivation for this study.

The issue of novelty aside, the following points are written treating this as an application study:

* Insufficient experimentation: The authors use three toy datasets, two of which they simplify further by binarizing. They also arbitrarily split them into 3 clients. I would have expected a much more extensive test-bed, where FedNAM would perform comparably to some baselines while being more interpretable.
* Insufficient descriptions about the experimental set-up: It would be impossible to replicate the authors’ work based on the description in the intro of section 6, as it is vague: for example, what was the exact structure of the network used for each feature?

**Questions:**

Please see and address the weaknesses listed above.

---

### Official Review · Reviewer_owEF · 2024-10-29

**Soundness:** 2
**Presentation:** 2
**Contribution:** 2
**Rating:** 1
**Confidence:** 4

**Summary:**

This paper introduces a novel federated neural additive model, named FedNAM, designed to address the interpretability challenges in federated learning. To achieve this, the authors propose leveraging neural additive models (NAMs) within a federated learning framework. FedNAM's interpretability is demonstrated using three small datasets, including the UCI heart disease, Wine, and Iris, showing its potential to provide transparent insights into federated model predictions.

**Strengths:**

The target problem addressed by this paper is both interesting and crucial for advancing interpretability in federated learning. The proposed approach, FedNAM, is straightforward and easy to understand, making it accessible for practical implementation. Through the integration of neural additive models within a federated framework, this method offers a promising solution for enhancing model transparency across distributed datasets.

**Weaknesses:**

- The authors should improve the paper's structure and clarity, as some sections, like the two paragraphs in Section 2, appear redundant.
- The novelty of FedNAM is limited; it primarily extends existing NAMs to a federated learning framework.
- The experimental setup lacks essential details, including dataset statistics, client deployment specifics, and hyperparameters.
- Although interpretability is the primary goal, the authors should provide accuracy metrics for FedNAM and compare it against existing methods.
- Federated learning's characteristic heterogeneity is not adequately addressed; experiments under heterogeneous conditions would strengthen the study.
- The datasets used are quite small, which limits generalizability. Real-world applications often involve datasets with hundreds of features. Demonstrating FedNAM's effectiveness on larger datasets would enhance its practical relevance.

**Questions:**

See weaknesses.

---

### Official Review · Reviewer_VgQV · 2024-11-03

**Soundness:** 3
**Presentation:** 2
**Contribution:** 2
**Rating:** 5
**Confidence:** 4

**Summary:**

This paper studies the interpretability and explainability of federated learning. The federated version of the neural additive model (NAM) is proposed to obtain interntable analysis results. The proposed method is evaluated on three public datasets.

**Strengths:**

- Interpretability is an important topic.
- The motivation is clearly demonstrated.
- The method is easy to follow.

**Weaknesses:**

- The technical novelty is limited. The adaptation of the NAM into federated learning seems straightforward.
- The results of using deep neural networks should be included in experiments.
- There are no experimental results about the task performance.
- It would be better to interpret the results according to the client data distributions.
- It is not clear what are the practical benefits of applying this method in real-world applications such as medical data.

**Questions:**

Please see the weakness part.

---

### Note · Authors · 2024-11-23

**Comment:**

We need to revaluate our methodology by conducting more experimentation, so we are considering withdrawing the paper from submission.

**Withdrawal Confirmation:**

I have read and agree with the venue's withdrawal policy on behalf of myself and my co-authors.